# Research on 3D point cloud alignment algorithm based on SHOT features

**Zheng Fu** [ID], **Enzhong Zhang** [ID]*, **Ruiyang Sun, Jiaran Zang, Wei Zhang**

School of Mechatronical Engineering, Changchun University of Technology, Changchun, China

* znz612@sina.com

**Data Availability Statement:** All relevant data are within the manuscript and its Supporting information files.

**Funding:** This paper is supported by Natural Science Foundation of Jilin Province. Project name:

## Abstract

To overcome the problem of the high initial position of the point cloud required by the traditional Iterative Closest Point (ICP) algorithm, in this paper, we propose a point cloud registration method based on normal vector and directional histogram features (SHOT). Firstly, a hybrid filtering method based on the voxel idea is proposed and verified using the measured point cloud data, and the noise removal rates of 97.5%, 97.8%, and 93.8% are obtained. Secondly, in terms of feature point extraction, the original algorithm is optimized, and the optimized algorithm can better extract the missing part of the point cloud. Finally, a fine alignment method based on normal vector and directional histogram features (SHOT) is proposed, and the improved algorithm is compared with the existing algorithm. Taking the Stanford University point cloud data and the self-measured point cloud data as examples, the plotted iteration-error plots can be concluded that the improved method can reduce the number of iterations by 40.23% and 37.62%, respectively.

## Introduction

With the development of 3D point cloud scanning technology in recent years, its technology has been applied to many fields due to the advantages of the fast acquisition of 3D point cloud spatial data, high resolution, and high accuracy, such as 3D modeling [1, 2], medical imaging [3], unmanned driving [4, 5], reverse engineering [6, 7], etc. The purpose of point cloud alignment is to solve the transformation matrix of point clouds of different poses at the same coordinates, and then use the transformation matrix to achieve accurate alignment of multi-view scans and finally obtain the complete model and scene. The key to point cloud processing technology lies in the alignment of the point cloud, and the quality of point cloud alignment directly affects the subsequent work. The higher the accuracy of alignment, the better the final result of obtaining the complete model; the faster the speed of alignment, the more beneficial to the development and application of alignment technology.

Regarding the point cloud alignment problem, many researchers have proposed various algorithms, among which the ICP algorithm [8] is the most widely used and the most classical one. The core of this algorithm is to iterate continuously to find the set of corresponding point pairs and calculate the optimal parameters of the rigid body transformation between the corresponding points to align the two point clouds. However, the algorithm suffers from the drawbacks of slow convergence and a tendency to bind to locally optimal solutions. To address

Research on a new method of composite free-form surface robot airbag polishing. The project number is 20230101336JC.

**Competing interests:** 作者已声明不存在竞争利益。

these drawbacks, researchers started to improve the ICP algorithm. Yao Zongwei et al [9] proposed an enhanced ICP algorithm based on curvilinear similarities to improve alignment speed and accuracy in unstructured environments using curvilinear similarities. Jiao-Long Yang et al [10] proposed the first globally optimal ICP (Go-ICP) algorithm, which derives new upper and lower bounds for the error function by searching the entire 3D motion space, speeding up the operation while ensuring global optimality.B. Sofien et al [11] formulated the alignment optimization by using sparse induced parametrization for ICP algorithm by using sparse induced parametrization to formulate alignment optimization, dealing with outliers and incomplete data to achieve excellent alignment results. S. James et al [12] proposed (Generalized ICP, G-ICP), using information channels integrated into the point covariance weights used. J. Serafin et al [13] proposed an algorithm for determining the data association between two scatterplots based on the 3D shape around the scatterplots, using a least-squares formulation of the matching problem, minimizing the error measure based on surface characteristics (curvature, normal). Feature-based matching methods are also a more common class of point cloud alignment algorithms, where the key is to find the distinct points of features and the exact correspondence. A pose estimation algorithm for the pose-picking task using point cloud data was proposed by Li Mingyu et al [14], using Curve Set Feature (CSF) to describe the points on the surface while the pose can be evaluated, and the matching combines the Point Pair Feature (PPF) algorithm in 2D space with the nearest The Rotating Matching Feature (RMF) is proposed to effectively match the descriptor curve set feature (CSF) when matching by combining the matching idea of Point Pair Feature PPF in 2D space with Nearest Neighbor Search, which proves to be robust to noise. Xiong Fu et al [15] proposed a covariance descriptor-based feature matching method, which greatly reduces the matching time and has better robustness to rigid transformations and noise by computing the covariance matrix of geometric features of feature points and their neighborhood points. Yan Lei et al [16] proposed a global optimization algorithm based on an attitude map to optimize the point cloud trajectory by attitude map and use the optimized trajectory for point cloud fine alignment, which reduces the alignment error. There are some other types of point cloud alignment optimization algorithms. Jun Zhang et al [17] introduced the implementation of iterative node placement for non-rigid alignment of radial basis functions to reduce the alignment error adaptively. Qinhua Tian et al [18] proposed a point cloud alignment method based on SHOT feature fusion, which uniquely describes key points by improving SHOT feature descriptors and solves the corresponding points in the point cloud using the minimum variance method, and the accuracy and speed of alignment are improved.

Currently, most point cloud alignment algorithms are divided into two parts: coarse alignment and fine alignment. The coarse alignment process is to provide a good initial pose for the subsequent fine alignment operation by finding the transformation matrix between two point clouds; the fine alignment is to iterate based on the coarse alignment until the best transformation is obtained.

## Coarse matching based on SHOT and ISS algorithms

The process of point cloud coarse alignment includes point cloud filtering, point cloud feature point extraction, feature point matching, mismatched pair rejection, etc. The general flow is shown in (Fig 1):

### Hybrid filtering based on voxel idea

When acquiring point cloud data, some noise points will inevitably appear in the point cloud data due to the influence of the accuracy of the equipment, operator experience, environment,

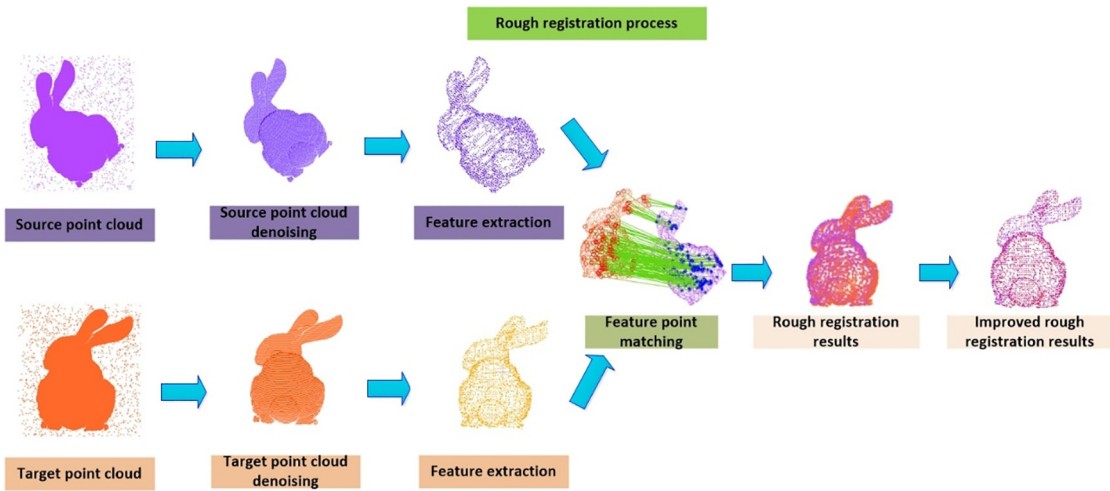

**Fig 1. Point cloud coarse alignment process.**

and other factors. As the first step in the coarse alignment process, the filtering effect will directly affect the subsequent alignment accuracy.

The original single statistical and radius filtering methods both have the problem that some noise points cannot be removed, which will affect the later point cloud alignment, so this paper proposes a hybrid filtering method based on the voxel idea. The voxel can grid the point cloud space, and reduce the number of points while keeping the shape characteristics of the point cloud unchanged, while also retaining the basic spatial structure information, which makes the accuracy and efficiency of hybrid filtering greatly improved. The specific steps are as follows:

1. Create an enclosing box for the original point cloud with the volume of $v$ and the length, width, and height of $l$, respectively, i.e. $l \times w \times h$;

2. Calculate the rational voxel edge length $L$; divide $M$ voxels according to the voxel edge length $L$ with the following expression:

$$M = \frac{v}{L^3} \qquad (1)$$

Here the downsampling rate $\rho$ is combined with the point cloud density to determine the voxel edge length $L$, i.e.

$$L^{'} = s\sqrt[3]{\frac{1}{\gamma}} \qquad (2)$$

Where: $L'$ is the edge length of the voxel to be determined; $S$ is the average number of point clouds per voxel; $\gamma$ is the point cloud density. $S$ The relationship between $\gamma$ and is

$$s = \frac{1}{\rho} \qquad (3)$$

$$\gamma = \frac{N}{v} \qquad (4)$$

Where: $N$ is the total number of point clouds. The Eqs (3) and (4) are treated into (2) to obtain

$$L' = \frac{1}{\rho} \sqrt[3]{\frac{v}{N}} \tag{5}$$

3. Determine the number of voxel cells. Divide the point cloud data into $M'$ voxel cells according to the voxel edge length to be determined $L'$:

$$M' = \frac{v}{L'^3} = \rho^3 N \tag{6}$$

4. Find all voxels containing point clouds and calculate the number of points within each voxel to find the average density of voxel cells with edge length $L'\bar{\gamma} = \frac{N}{m}$, when $\bar{\gamma} < s$ $L' = L' + \lambda$; when $\bar{\gamma} > s$, $L' = L' - \lambda$; where $\lambda$ is the growth factor, $\lambda = \sqrt[3]{\frac{v}{N}}$; stopping when $\bar{\gamma} = s$, and finally $L = L'$.

5. Divide the enclosing box of the point cloud $P$ by iterating the edge length of the voxel $L$ for several times, and calculate the center of gravity of all point cloud data within the voxel, and use the center of gravity point $P'_i$ as the downsampling point, whose expression is

$$P'_i = \frac{\sum_{j=1}^{N_{m_i}} (x_j, y_j, z_j)}{N_{m_i}} \tag{7}$$

Where: $N_{m_i}$ is the number of points in the voxel of $i$ and $i \in [1, 2, \ldots, m]$. The new point cloud $P'$, which is the sampled point cloud, is calculated from Eq (7).

6. Combine statistical filtering and radius filtering to remove the noise. Iterate through all the point cloud data after sampling, remove the noise, and the filtering is finished.

In this paper, the Stanford Bunny point cloud dataset (https://www.cc.gatech.edu/projects/large_models/bunny.html) and the self-measured free-form dataset are used for point cloud filtering verification. 5000 random noise points are added to the three-point cloud datasets respectively, and the difference between the single filtering algorithm and the algorithm in this paper can be seen by the filtering results, the effect comparison in (Fig 2):

As can be seen from Table 1 above, the noise removal rates of the filtering methods in this paper are 97.5%, 93.8%, and 97.8%, respectively, and the filtering effect has been improved and the point cloud data features have been well preserved.

## Point cloud feature point extraction

The extraction of feature points from point cloud data can also affect the subsequent alignment work. The single feature point extraction method has some shortcomings, such as not being able to retain the local features of the point cloud, missing internal points of the point cloud, low extraction efficiency, etc. The original point cloud data and the single feature point extraction results are shown in (Figs 3 and 4), respectively. In this paper, voxel downsampling and internal shape descriptor (ISS) feature point extraction methods are combined to extract feature points of point clouds. Voxel downsampling is a more efficient method to process point cloud data. In this paper, we first use the voxel downsampling method to streamline the point cloud data for the first time to save the important data such as shape and geometric features of

(a)

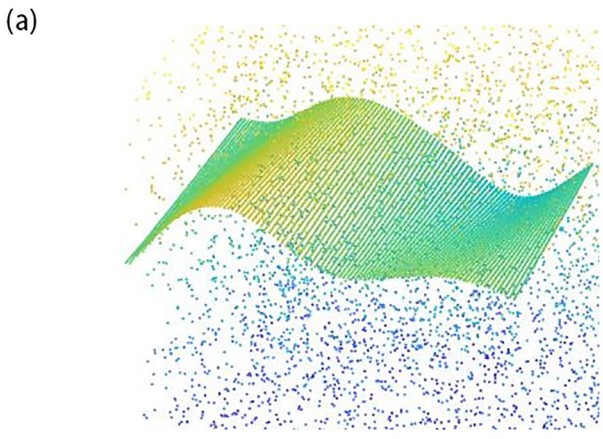

(b)

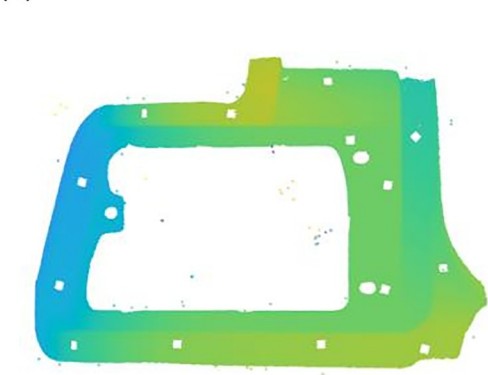

(c)

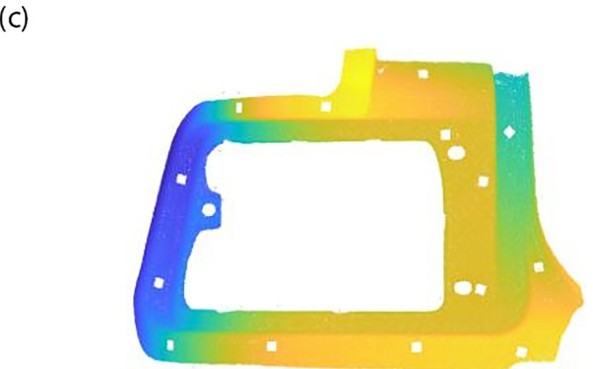

**Fig 2. Effect comparison chart.** (a) Free-form data after noise addition. (b) Single filtering. (c) This paper filtering.

the point cloud, and then use the ISS extraction feature point method to further save the surrounding data, so that the local features of the point cloud data can be better preserved. The sampling results of the Stanford rabbit using the method in this paper are shown in (Fig 5). The specific steps after improvement are as follows:

1. Set the voxel size. Continued downsampling of the filtered point cloud using voxels;

2. Set the search radius. Set the search radius $r$ for each query point $P_i$.

**Table 1. Number of noise points removed by different filters.**

| | Filtering Method | Total number of noise point clouds | Number of filtered noise points | Filtration rate |
|---|---|---|---|---|
| **bun090** | Single filtering method | 35379 | 4724 | 94.48% |
| | The filtering method in this paper | | 4875 | 97.5% |
| **Free-form surface** | Single filtering method | 26316 | 4496 | 89.92% |
| | The filtering method in this paper | | 4690 | 93.8% |
| **Auto Parts** | Single filtering method | 165790 | 4779 | 95.6% |
| | The filtering method in this paper | | 4892 | 97.8% |

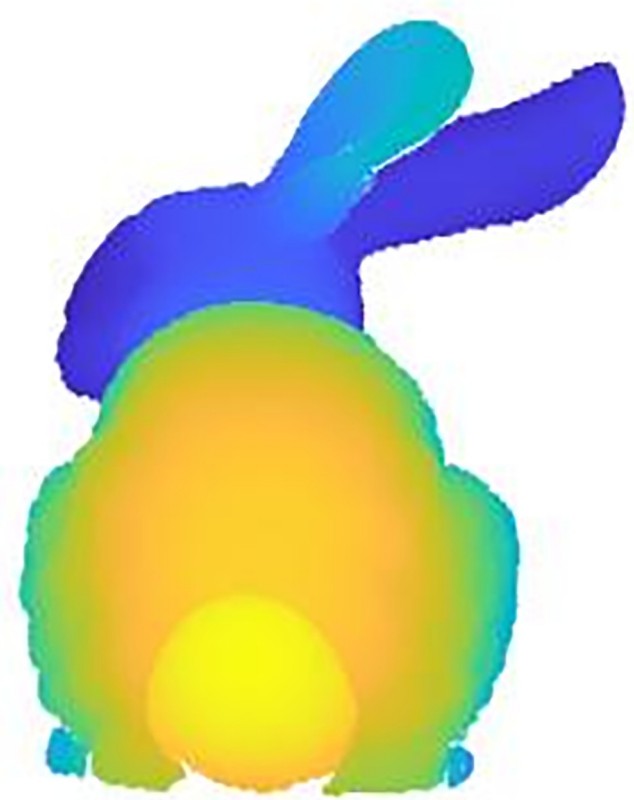

**Fig 3. Original point cloud data.**

3. Calculate the Euclidean distance between each query point $P_i$ and the points in the neighborhood and set the weights $w_{ij}$;

$$w_{ij} = \frac{1}{\|P_i - P_j\|} \ \|P_i - P_j\| < r \tag{8}$$

4. Calculate the covariance matrix of each query point $P_i$ with all points in the neighborhood $COV(P_i)$, which is calculated as follows:

$$COV(P_i) = \frac{\sum_{\|P_i - P_j\| < r} w_{ij}(P_i - P_j)(P_i - P_j)^T}{\sum_{\|P_i - P_j\| < r} w_{ij}} \tag{9}$$

5. Calculate the eigenvalues of all the covariance matrices $COV(P_i)$ in 4), denoted as $\{\lambda_i^1, \lambda_i^2, \lambda_i^3\}$, and rank the eigenvalues from largest to smallest;

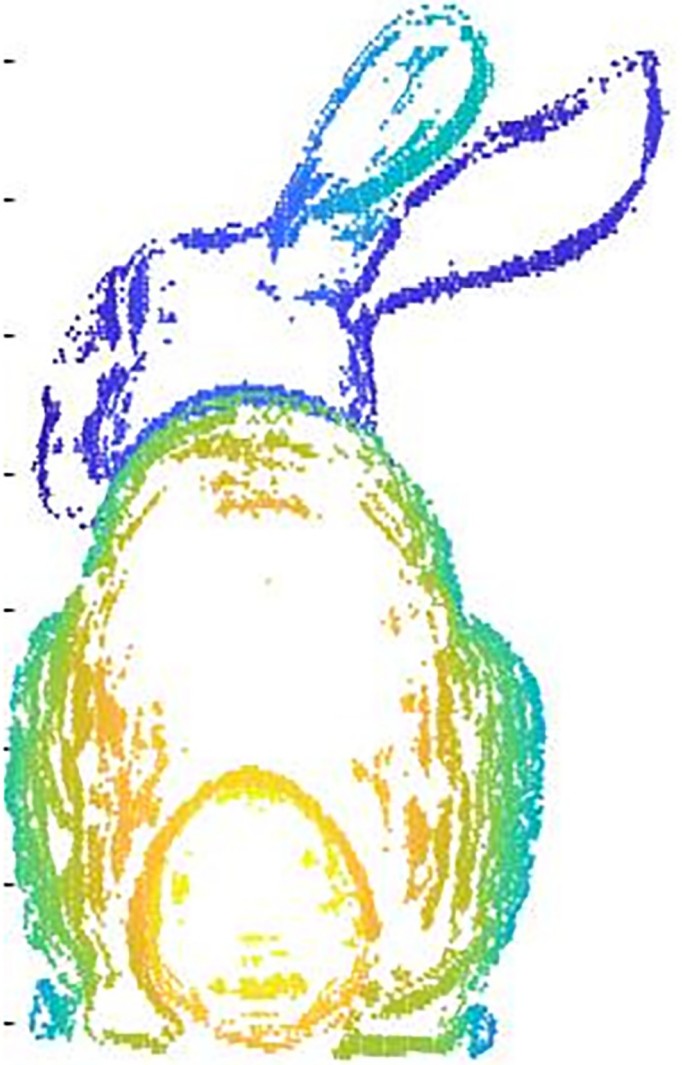

**Fig 4. Original method (deficiency).**

6. Set the threshold value. Set the threshold values of $\varepsilon_1$ and $\varepsilon_2$, which are ISS feature points when Eq (9) is satisfied.

$$\frac{\lambda_i^2}{\lambda_i^1} \leq \varepsilon_1$$
$$\frac{\lambda_i^3}{\lambda_i^2} \leq \varepsilon_2 \tag{10}$$

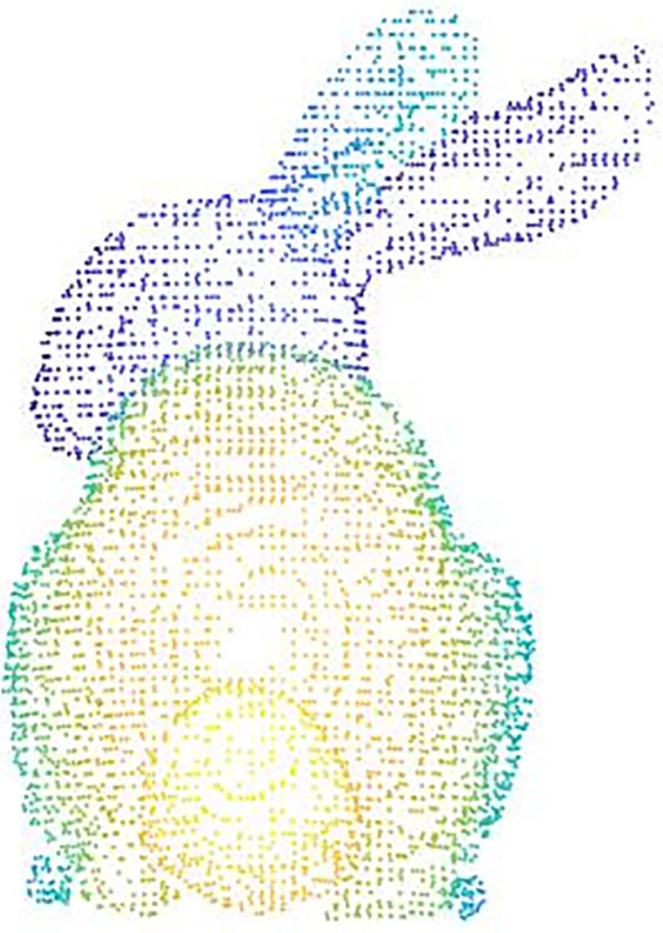

**Fig 5. Sampling by this method.**

## Point cloud SHOT feature matching

After the feature points of point cloud data are determined, the establishment of pre-matching pairs is carried out next. The basic idea is to establish a local coordinate system based on the neighborhood points, so the neighborhood space is divided into several subspaces; then the normal vector features of all points in the subspaces are organized into histograms and encoded; finally, the histograms of each subspace are combined, and then the 3D descriptors are obtained. The specific calculation is as follows.

1. For each query point $P_i$ calculate and construct the point neighborhood covariance matrix $Q$, which is given by

$$Q = \frac{\sum_{d_k \leq r}(r - d_k)(P_k - \hat{P})(P_k - \hat{P})^T}{\sum_{d_k \leq r}(r - d_k)} \tag{11}$$

Where: $r$ represents the radius of the neighborhood; $P_k$ is each point in the neighborhood;

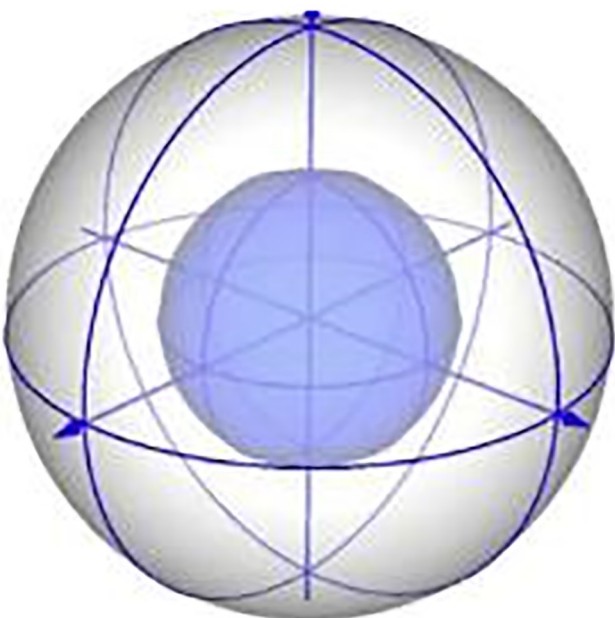

**Fig 6. SHOT feature division.**

$\hat{P}$ represents the center of mass of all points in the neighborhood; $d_k$ is the distance between the points $\hat{P}$ in the neighborhood and the center of mass.

2. Solve the covariance matrix $Q$ to obtain the eigenvalues and their corresponding eigenvectors, and sort the eigenvalues from largest to smallest $\lambda_1$, $\lambda_2$, $\lambda_3$; the corresponding eigenvectors correspond to the three axes of $x$, $y$, $Z$: $u_1$, $u_2$, $u_3$.

3. Create a spherical neighborhood with $P_i$ as the center and a radius of $r$, and divide the spherical neighborhood into 32 spaces as shown in (Fig 6).

4. Calculate the normal vectors of the neighboring points $P_k$ and the query point $P_i$ in the subspace, respectively, and find the cosine of the angle between them with the formula

$$COS\theta_{ni} = u_{ni} \cdot u_3 \tag{12}$$

Where: $u_{ni}$ denotes the normal vector of the $i$ point in the $n$ subspace; $u_3$ is the normal vector of the query point $P_i$, i.e., the axis.

5. Count all the values of the cosine of the pinch angle calculated from Eq (12) to build a high-dimensional histogram feature. The visual feature matching is shown in (Fig 7):

## Point cloud mismatch rejection

After the above-mentioned series of processing, most of the noise points in the point cloud data have been removed, but it does not reach the ideal state, and the unremoved noise points and the feature points at the edges will form a certain chance of mismatching pairs. Therefore, this paper proposes to perform false match rejection in two stages, firstly, in the feature detection stage, we use the edge point detection algorithm to eliminate the feature points at the edge

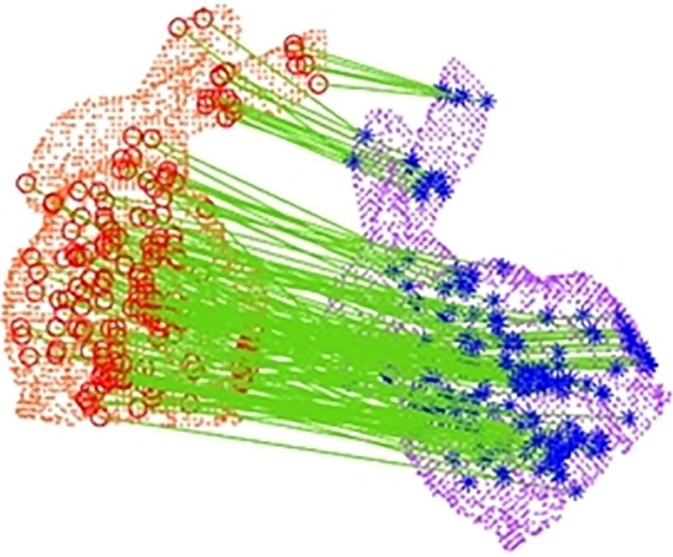

**Fig 7. Visual feature matching.**

of the point cloud data based on this algorithm, and then get the feature points with high quality, which reduces the false match pairs generated by the edge feature points; then, after the point cloud feature matching, we propose the k-means clustering [19] and RANSAC algorithm based on the combination of Then after the point cloud features are matched, a combination of k-means clustering and RANSAC algorithm is proposed to eliminate the false match pairs effectively and quickly.

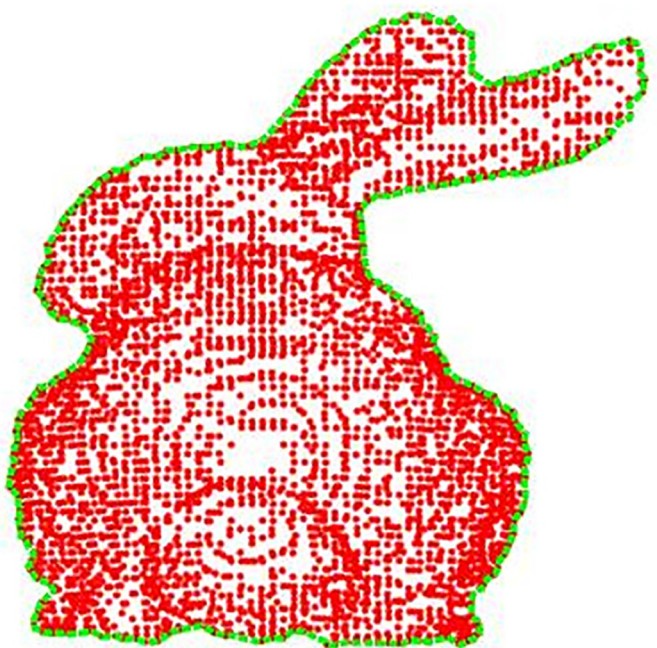

**Fig 8. Point cloud edge feature extraction.**

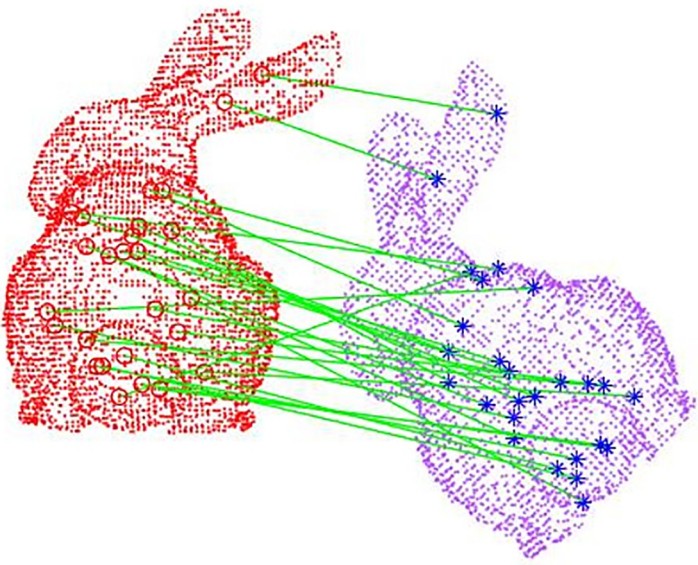

**Fig 9. After false match rejection.**

In the edge point rejection method for point cloud data, edge point detection is at the core. It uses kd-tree to search all the nearest neighbor points in the area, traversing all the point cloud data, fitting the nearest objects to a surface, and using the projection method to calculate the angle between the point and the projection point. If it is above the set threshold, the point is recognized as an edge point, and vice versa, the point is skipped, and all the edge points are connected and displayed using MATLAB, and the results are shown in (Fig 8):

After eliminating the edge feature points of the point cloud data, there are still some mismatching pairs in the point cloud feature matching, so the next step is to eliminate the point cloud mismatching pairs. Firstly, we introduce the k-means idea to cluster the matching pairs and keep the clusters with the largest number of matching pairs after the division, to eliminate some of the mismatched pairs. Then, based on the majority of correct matching pairs, we adopt the RANSAC algorithm, which applies randomness to randomly sample the feature points and randomly select $m$ feature matching pairs to calculate the transformation relationship between the original point cloud and the target point cloud. The above process is repeated, and the transformation relationship with the largest feature-matching pair is taken as the correct hypothesis, and the feature-matching pairs that satisfy the transformation relationship are retained, and the feature matching pairs that do not satisfy the transformation relationship are eliminated, as shown in (Fig 9).

## Fine alignment based on normal vector and SHOT features

After the coarse alignment of the point cloud, the original point cloud $P$ and the target point cloud $Q$ have been roughly aligned. To further improve the matching accuracy, the two-point clouds need to be finely aligned. The iterative closest point algorithm (ICP) is the most common method for fine alignment, but this algorithm has the defects of large computation, long matching time, and easy falling into local optimum. Therefore, in the case of obtaining a good initial position by coarse alignment, this paper proposes an ICP fine alignment method based on normal vector and SHOT features. The procedure is as follows.

1. The voxel grid method and the ISS feature point combination process are used to sample the point cloud, and the normal vector-based idea is used to obtain the point cloud feature point set, which effectively reduces the computational complexity.

2. The introduction of kd-tree to search the nearest neighbor points of $P_i$ greatly improves the search efficiency.

3. Calculate the SHOT features of $P_i$ and the nearest neighboring points, and use the SHOT features between point pairs to determine the unique point $Q_i$ corresponding to form the set of corresponding points, thus increasing the number of correct match pairs and the improvement in accuracy.

4. Use SVD to solve the rigid body transformation matrix for several iterations until the error condition is satisfied, the iteration is terminated and the alignment is completed.

Although the improved algorithm in this paper increases the computation of SHOT features, the whole process, the idea based on normal vectors, and the method of using kd-tree to search for nearest neighbor points both save time greatly, so the alignment efficiency does not decrease and because the use of point cloud feature descriptors makes the number of correct matching pairs of point clouds increase, effectively improving the alignment accuracy.

## Simulation analysis

To verify the feasibility of the proposed method in this paper, the original ICP [8] algorithm, Kd-tree accelerated ICP algorithm [20], and PL-ICP (Point-to-line ICP) algorithm [21] are compared experimentally with the method in this paper as a way to verify the robustness of the method in this paper. This paper uses MATLAB R2022b and PCL programming library with Intel(r)core(TM) i5-11400h @ 2.70 GHz CPU and 16G memory.

The point cloud data utilized in this paper was obtained from the Stanford University point cloud database and a free-form dataset that was machined and measured in-house, with the machining and measurement equipment shown in (Fig 10):

The measurement part of the information acquisition device uses Keene's contact digital sensor, model GT2-H50, with an accuracy of 3.5 $\mu m$, a resolution of 0.5 $\mu m$, and a bow trajectory for measurement with a 1mm interval between measurement points. The experiment uses a self-developed CNC machine with an X-axis range of 0-270mm, a Y-axis range of 0-245mm, and a Z-axis range of 0-225mm.

### Coarse alignment simulation analysis

The improved coarse alignment algorithm in this paper is compared and analyzed with the existing algorithms mentioned above, and the root mean square error (RMSE) is used as the evaluation index, which is defined by the following formula:

$$RMSE = \sqrt{\frac{\sum_{i}^{N} \|Tp_i - q_i\|^2}{N}} \tag{13}$$

Where $q_i$ is the nearest point in the target point cloud to the point $p_i$ in the source point cloud, $N$ is the number of point clouds, and $T$ is the translation matrix.

In this paper, the improved alignment method has better accuracy and robustness compared with the existing methods for the alignment of the Bunny point cloud dataset and the

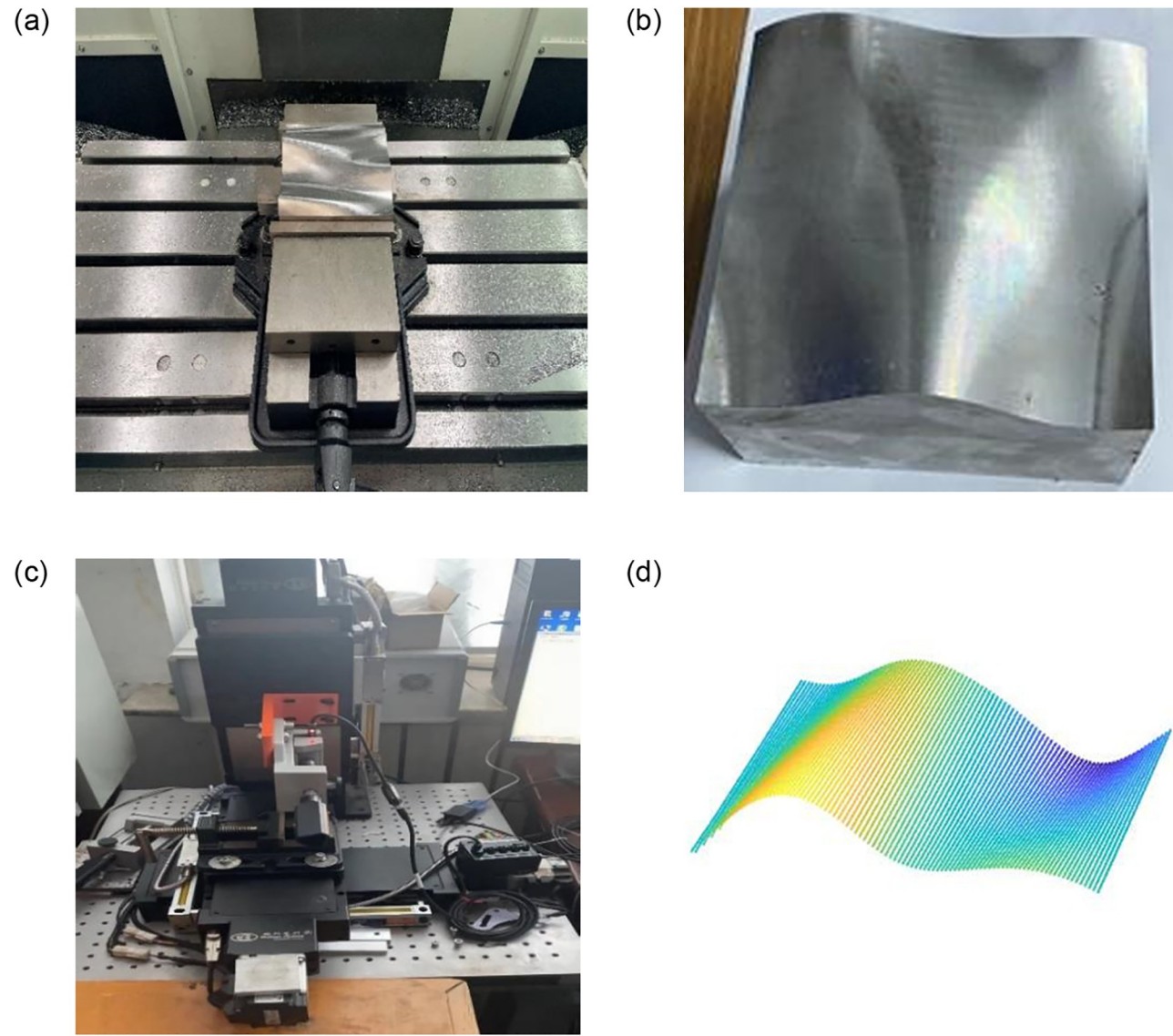

**Fig 10. Processing and measuring equipment.** (a) Free-form machining. (b) Workpiece model. (c) Free-form surface measurement. (d) Self-measured point cloud data.

self-measured free-form surface point cloud dataset. The improved alignment method is more efficient, and the rough alignment results are shown in Table 2.

### Precise matching accuracy simulation analysis

To verify the accuracy of the fine alignment in this paper, three existing algorithms are compared and analyzed with this paper's method, and the alignment accuracy is selected as RMSE, and the results are shown in (Fig 11). When the number of iterations is 20, the error generated by this method is reduced by 29.97% and 33.16% compared with the original ICP algorithm; when the algorithm error value is $3.8612 \times 10^{-6}$ $m$, the number of iterations required by this method is reduced by 40.23% and 37.62% compared with the ICP algorithm. When the error

**Table 2. Analysis of the results of the rough alignment experiments.**

| Alignment Method | Number of iterations | RMSE / m | Number of iterations | RMSE / m |
|---|---|---|---|---|
| ICP | 10 (Stanford data) | $9.5236 \times 10^{-6}$ | 10 (Free-form surface) | $8.796 \times 10^{-6}$ |
| ICP − Kd − tree | | $7.3869 \times 10^{-6}$ | | $7.3926 \times 10^{-6}$ |
| PL − ICP | | $6.6023 \times 10^{-6}$ | | $6.938 \times 10^{-6}$ |
| Methodology of this article | | $5.5324 \times 10^{-6}$ | | $5.938 \times 10^{-6}$ |
| ICP | 25 (Stanford data) | $5.1328 \times 10^{-6}$ | 25 (Free-form surface) | $3.617 \times 10^{-6}$ |
| ICP − Kd − tree | | $4.5903 \times 10^{-6}$ | | $3.597 \times 10^{-6}$ |
| PL − ICP | | $4.5366 \times 10^{-6}$ | | $3.5391 \times 10^{-6}$ |
| Methodology of this article | | $4.4796 \times 10^{-6}$ | | $3.4796 \times 10^{-6}$ |

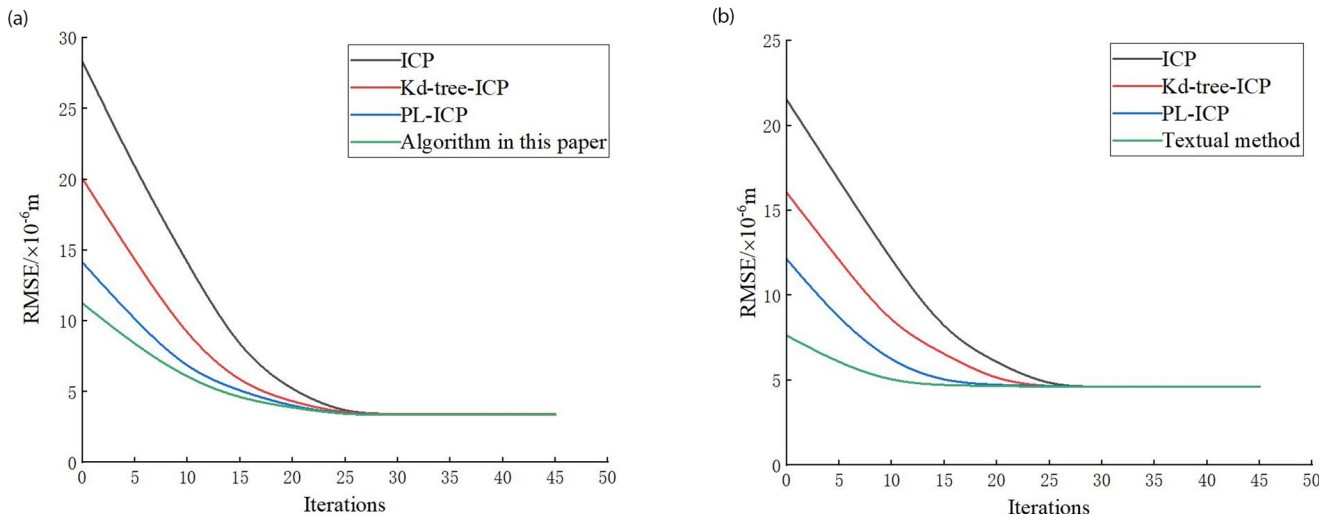

**Fig 11. Precision alignment results.** (a) Bunny point cloud data. (b) Free-form surface.

value of the algorithm is, the number of iterations required by this method is reduced by 40.23% and 37.62% compared with the ICP algorithm.

## Conclusions

In this article, a new point cloud alignment method is proposed. By combining the improved coarse alignment algorithm and the ICP fine alignment algorithm based on normal vector and SHOT features, a better alignment effect is achieved. Compared with the traditional ICP algorithm, the method in this paper has better robustness in 3D point cloud alignment and the matching efficiency is improved. The conclusions of the research in this paper are as follows:

1. In the coarse alignment stage, the improved algorithm is verified. Under the improved point cloud filtering and point cloud feature point extraction processing, the noise rejection rates in the point cloud data are 97.5%, 97.8%, and 93.8%, respectively, which are better.

2. In the fine alignment stage, compared with the traditional ICP algorithm, the method in this paper has better description capability and robustness. Experimental verification of the fine alignment method in this paper shows that the improved algorithm reduces the

number of iterations by 40.23% and 37.62%, respectively, and the alignment accuracy and matching speed are significantly improved.

## Supporting information

**S1 Data.**
(ZIP)

## Acknowledgments

The authors would like to thank the anonymous reviewers and the editor for their valuable comments and suggestions for the improvement of this paper.

## Author Contributions

**Data curation:** Ruiyang Sun.

**Software:** Wei Zhang.

**Supervision:** Jiaran Zang.

**Writing – original draft:** Zheng Fu.

**Writing – review & editing:** Enzhong Zhang.

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
