## [Decision Letter · Decision Letter 0]

6 Nov 2023

PONE-D-23-29806Research on 3D point cloud alignment algorithm based on SHOT featuresPLOS ONE

Dear Dr. Zhang,

Thank you for submitting your manuscript to PLOS ONE. After careful consideration, we feel that it has merit but does not fully meet PLOS ONE’s publication criteria as it currently stands. Therefore, we invite you to submit a revised version of the manuscript that addresses the points raised during the review process.

We look forward to receiving your revised manuscript.

Kind regards,

Xuejian Wu, Ph.D.

Academic Editor

PLOS ONE

Journal Requirements:

Software, J.Z., W.Z.; data curation, R.S; writing—original draft preparation, Z.F.; writing—review and editing, E.Z.; All authors have read and agreed to the published version of the manuscript.

Reviewers' comments:

Reviewer's Responses to Questions

**Comments to the Author**

1. Is the manuscript technically sound, and do the data support the conclusions?

Reviewer #1: Yes

2. Has the statistical analysis been performed appropriately and rigorously? 

Reviewer #1: Yes

3. Have the authors made all data underlying the findings in their manuscript fully available?

Reviewer #1: Yes

4. Is the manuscript presented in an intelligible fashion and written in standard English?

Reviewer #1: Yes

5. Review Comments to the Author

Reviewer #1: The paper titled "Research on 3D point cloud alignment algorithm based on SHOT features" by Zheng Fu et al introduces a new method for aligning 3D point clouds. It combines an enhanced coarse alignment algorithm with the ICP fine alignment algorithm using normal vectors and SHOT features to achieve superior alignment results. Compared to the traditional ICP algorithm, this method offers better robustness and improved matching efficiency.

The paper is properly structured and well written. The prosed point cloud alignment can be useful in contain fields. Therefore the paper in my opinion certainly should be considered for publication in PLOS ONE. However, there are some typos throughout the paper. To improve overall clarity and readability, the authors should address the comments below to eliminate confusion and enhance the overall readability.

1. In the abstract, on line 3, the acronym should be placed in parentheses as follows: "… method based on the Signature of Histogram of Orientation (SHOT) feature is proposed …."

2. In the abstract, on line 9, the mention of "directional histogram features (SHOT)" is confusing. Revising it as "…on normal vector and SHOT features is proposed,…" might be better.

3. On page 12, please add a reference or web link for the Stanford Bunny dataset.

6. PLOS authors have the option to publish the peer review history of their article (what does this mean?). If published, this will include your full peer review and any attached files.

Reviewer #1: **Yes: **Weicheng Zhong

---

## [Author Response · Author response to Decision Letter 0]

12 Dec 2023

Response to reviewer

Dear Editors and Reviewers:

 We gratefully thank the editor and all reviewers for their time spend making their constructive remarks and useful suggestions, which has significantly raised the quality of the manuscript and has enable us to improve the manuscript. Each suggested revision and comment, brought forward by the reviewers was accurately incorporated and considered. Below the comments of the reviewers are response point by point-and the revisions are indicated.

Responds to the reviewer’s comments:

Reviewer 1:

1. In the abstract, on line 3, the acronym should be placed in parentheses as follows: "… method based on the Signature of Histogram of Orientation (SHOT) feature 

 is proposed …."

 The author's answer: ……directional histogram features (SHOT).

Reviewer 2：

2. In the abstract, on line 9, the mention of "directional histogram features (SHOT)" is confusing. Revising it as "…on normal vector and SHOT features is 

 proposed…" might be better.

 The author's answer: We propose a point cloud registration method based on normal vector and directional histogram features (SHOT).

Reviewer 3:

3. On page 12, please add a reference or web link for the Stanford Bunny dataset.

 The author's answer: In this paper, the Stanford Bunny point cloud dataset (https: //www. cc. gatech. edu/ projects/ large_models/ bunny. html) and the self-measured free-form dataset are used for point cloud filtering verification.

4. I make sure to cite all the figures and tables in the article, including Figures 2, 7, 10, 11, 12 and 13, Table 2.

---

## [Editor Report · Decision Letter 1]

18 Dec 2023

Research on 3D point cloud alignment algorithm based on SHOT features

PONE-D-23-29806R1

Dear Dr. Zhang,

We’re pleased to inform you that your manuscript has been judged scientifically suitable for publication and will be formally accepted for publication once it meets all outstanding technical requirements.

Kind regards,

Xuejian Wu, Ph.D.

Academic Editor

PLOS ONE
---

## [Editor Report · Acceptance letter]

9 Jan 2024

PONE-D-23-29806R1 

PLOS ONE

Dear Dr. Zhang, 

I'm pleased to inform you that your manuscript has been deemed suitable for publication in PLOS ONE. Congratulations! Your manuscript is now being handed over to our production team.

Kind regards, 

on behalf of

Dr. Xuejian Wu 

Academic Editor

PLOS ONE